# Disease-modifying therapies and features linked to treatment response in type 1 diabetes prevention: a systematic review

Jamie L. Felton[1,2], Kurt J. Griffin[3,4], Richard A. Oram[5,6,7], Cate Speake [8], S. Alice Long [9],
Suna Onengut-Gumuscu [10], Stephen S. Rich [10], Gabriela S. F. Monaco[1,2], Carmella Evans-Molina[1,11],
Linda A. DiMeglio [1,2], Heba M. Ismail[1], Andrea K. Steck[12], Dana Dabelea[13], Randi K. Johnson[14,15],
Marzhan Urazbayeva[16], Stephen Gitelman[17], John M. Wentworth[18,19], Maria J. Redondo[16,20,203],
Emily K. Sims [1,2,203✉] & ADA/EASD PMDI*

## Abstract

**Background** Type 1 diabetes (T1D) results from immune-mediated destruction of insulin-producing beta cells. Prevention efforts have focused on immune modulation and supporting beta cell health before or around diagnosis; however, heterogeneity in disease progression and therapy response has limited translation to clinical practice, highlighting the need for precision medicine approaches to T1D disease modification.

**Methods** To understand the state of knowledge in this area, we performed a systematic review of randomized-controlled trials with ≥50 participants cataloged in PubMed or Embase from the past 25 years testing T1D disease-modifying therapies and/or identifying features linked to treatment response, analyzing bias using a Cochrane-risk-of-bias instrument.

**Results** We identify and summarize 75 manuscripts, 15 describing 11 prevention trials for individuals with increased risk for T1D, and 60 describing treatments aimed at preventing beta cell loss at disease onset. Seventeen interventions, mostly immunotherapies, show benefit compared to placebo (only two prior to T1D onset). Fifty-seven studies employ precision analyses to assess features linked to treatment response. Age, beta cell function measures, and immune phenotypes are most frequently tested. However, analyses are typically not prespecified, with inconsistent methods of reporting, and tend to report positive findings.

**Conclusions** While the quality of prevention and intervention trials is overall high, the low quality of precision analyses makes it difficult to draw meaningful conclusions that inform clinical practice. To facilitate precision medicine approaches to T1D prevention, considerations for future precision studies include the incorporation of uniform outcome measures, reproducible biomarkers, and prespecified, fully powered precision analyses into future trial design.

## Plain language summary

Type 1 diabetes (T1D) is a condition that results from the destruction of a type of cell in the pancreas that produces the hormone insulin, leading to lifelong dependence on insulin injections. T1D prevention remains a challenging goal, largely due to the immense variability in disease processes and progression. Therapies tested to date in medical research settings (clinical trials) work only in a subset of individuals, highlighting the need for more tailored prevention approaches. We reviewed clinical trials of therapies targeting the disease process in T1D. While the overall quality of trials was high, studies testing individual features affecting responses to treatments were low. This review reveals an important need to carefully plan high-quality analyses of features that affect treatment response in T1D, to ensure that tailored approaches may one day be applied to clinical practice.

A full list of author affiliations appears at the end of the paper.

Type 1 diabetes (T1D) results from immune-mediated destruction of pancreatic beta cells[1]. Since the discovery of insulin over a century ago, treatment options for persons with type 1 diabetes (T1D) have shown remarkable advancements, including improved insulin formulations, delivery methods, and tools to monitor glycemia[2]. Even with these transformative advances, considerable negative impacts remain on health outcomes and quality of life[3–5]. In contrast, effective disease-modifying therapies aimed at the preservation of endogenous insulin production could not only improve these outcomes but also, if given early enough in the disease course, prevent the need for insulin replacement[6–9]. Because T1D is an autoimmune disease, many agents tested as potential disease-modifying therapies are immunomodulatory, while others target pathologic contributors such as glucose toxicity and beta cell health and function[10]. In 2022, the US Food and Drug Administration approved teplizumab, a monoclonal antibody targeting CD3, as the first therapy to delay the onset of clinical T1D in at-risk individuals[11].

The Precision Medicine in Diabetes Initiative (PMDI) was established in 2018 by the American Diabetes Association (ADA) in partnership with the European Association for the Study of Diabetes (EASD). The ADA/EASD PMDI includes global thought leaders in precision diabetes medicine who are working to address the burgeoning need for better diabetes prevention and care through precision medicine. This Systematic Review is written on behalf of the ADA/EASD PMDI as part of a comprehensive evidence evaluation of precision prevention in T1D in support of the 2nd International Consensus Report on Precision Diabetes Medicine[12]. The first ADA/ EASD Precision Medicine in Diabetes Consensus Report defined precision prevention as "using information about a person's unique biology, environment, and/or context to determine their likely responses to health interventions" and states that "precision prevention should optimize the prescription of health-enhancing interventions"[13]. Given that agents targeting these pathways may have potential adverse effects, and initial therapies may affect the efficacy and safety of subsequent treatment approaches, precision medicine is uniquely poised to identify which individuals stand to benefit the most from a given intervention and to optimize potential risk-benefit ratios for treated persons. Additionally, once further T1D disease-modifying therapies are approved for clinical use, precision medicine will facilitate the selection of therapies guided by the individual's disease, including potential combination regimens of disease-modifying therapies[14,15].

T1D development occurs along a spectrum of progressive beta cell destruction, beginning with loss of tolerance, reflected by the appearance of islet autoantibodies, and continuing with progressive hyperglycemia, abnormal glucose tolerance, and decline in endogenous insulin production, reflected by a decline in C-peptide[8]. Based on this, in 2015, the diabetes research community adopted a staging system, with the development of multiple islet autoantibodies now heralding Stage 1 T1D[9]. At the time of clinical T1D diagnosis, insulin replacement is required but endogenous insulin production, though diminished, can still be detected in most affected individuals. While an ideal goal is clinical T1D prevention, disease-modifying agents aimed throughout the spectrum of T1D progression have the potential to improve long-term outcomes[6,7]. Furthermore, given widely available participants and shorter total trial durations, agents planning to target earlier stages of the disease are often initially trialed in the new-onset period[16]. Therefore, we sought to understand the current state of knowledge regarding precision approaches to T1D disease modification, either to prevent the development of early-stage or clinical T1D (referred to as "prevention" studies) or to preserve endogenous insulin function around the time of clinical T1D diagnosis (referred to as "new-onset" studies). Specifically, we asked if individual characteristics have been robustly identified to select persons for therapeutic optimization of T1D disease-modifying therapies before or at the time of diagnosis. We reviewed and summarized existing trials in this area and identified individual characteristics associated with treatment effects.

## Methods

**Search strategy**. We developed a search strategy using an iterative process that involved Medical Subject Headings (MeSH) and text words. This search was refined based on a sensitivity check for key articles identified by members of the group (Supplementary Note 1). This strategy was applied to PubMed and EMBASE databases by librarians from Lund University on 2/22/2022.

**Systematic review**. The Covidence platform was utilized for stages of systematic review. To qualify for review, studies must have tested a disease-modifying treatment in either initially non-diabetic individuals at risk, or individuals with new-onset type 1 diabetes (within 1 year of diagnosis). Eligible study types included randomized controlled trials (RCTs); systematic reviews or meta-analyses of RCTs, or post hoc analyses of RCTs. Selected primary trials or longitudinal follow-up papers of primary trials had a total sample size ≥50 and were published as a full paper in English in a peer-reviewed journal within 25 years of the search (2/21/1997-2/22/2022). Papers focusing on a precision approach to identify features associated with a treatment response were also included if the total sample size was ≥10. Longitudinal follow-up papers of RCTs were included if they addressed follow-up data on time to diabetes, C-peptide area under the curve (AUC), or included "precision analyses" of specific individual features or measures of treatment response. Studies were excluded if they included mixed participant populations (i.e., type 1 and type 2 diabetes) or populations with inconsistent definitions across papers (i.e., latent autoimmune diabetes in adults). Several additional key articles previously known to the group of experts that also met inclusion criteria but were not included in the search results because of search restrictions designed to improve search feasibility were also included in the analysis. While systematic reviews and meta-analyses were included in the original search strategy to identify any existing meta-analyses aimed at precision approaches, none that met inclusion criteria were identified. All included papers were primary trials or post hoc analyses of primary trials.

Investigators independently screened and reviewed each potentially relevant article according to preliminary eligibility criteria determined by members of the review team. For Level 1 screening two investigators per article screened each title and abstract. Discordant assessments were discussed and resolved by consensus or arbitration after consultation with a member of the review leadership team (JLF, RO, KJG, MR, or EKS). For Level 2 screening of eligible articles, full texts were retrieved and reviewed by two independent reviewers using the inclusion/exclusion criteria. Discordant assessments were similarly discussed and resolved.

Two separate investigators per article extracted data from each article meeting inclusion criteria, with consensus determined by a member of the leadership team. Extracted data included study and publication name and date, if the study was single or multicenter, participant characteristics (age, sex, race, ethnicity, stage of disease), eligibility criteria, intervention details, details of metabolic monitoring, duration of follow-up, primary trial outcome, specific data on outcomes of intervention on time to diabetes (all available results) or C-peptide (at furthest reported

timepoint from treatment), and methods and findings surrounding precision analyses focused on treatment response.

The protocol for this review was registered (PROSPERO ID: CRD42022310063) before implementation and amended during review to edit group members and for feasibility, to add further exclusion criteria (populations defined as latent autoimmune diabetes and trials or follow-up studies with <50 participants).

**Risk of bias assessments.** Investigators also independently performed quality assessments using Covidence's Cochrane Risk of Bias template in tandem for each eligible study; this included assessments of sequence generation, allocation concealment, masking of participants/personnel, masking of outcome assessment, incomplete outcome data, selective reporting, and any other sources of bias to order to determine the overall risk of bias.

**Statistics and reproducibility.** Because of the heterogeneity of clinical interventions (e.g., agent tested, study design, analytical methodology, etc.), we were unable to perform a meta-analysis but instead completed summaries of relevant studies. A forest plot was generated using hazard ratios from all included prevention studies. No studies were missing data or required data conversion for this. Reproducibility was ensured by a dual investigator review of each article at each review stage.

**Reporting summary.** Further information on research design is available in the Nature Portfolio Reporting Summary linked to this article.

## Results

**Systematic review results.** From 1006 studies identified by PubMed and Embase searches, 75 were eligible for extraction (Fig. 1). This included original trial papers, trial longitudinal follow-up papers, and papers focused specifically on a precision analysis surrounding treatment response in prevention trials (15

papers from 11 prevention trial cohorts)[17–31] and in individuals with new-onset T1D (60 total papers from 45 new-onset trial cohorts)[32–91].

The 15 articles on T1D prevention generated from 11 trials are summarized in Tables 1 and 2. Primary prevention studies, conducted prior to autoantibody seroconversion in genetically at-risk individuals testing development of islet autoantibodies or time to T1D, comprised 27% (3/11) of trials; 63% (7/11) of trials were secondary prevention studies testing effects of interventions after seroconversion in autoantibody-positive individuals on time to T1D; one trial tested both genetically at-risk infants and autoantibody-positive siblings. Further inclusion criteria for trials included measures of beta cell function, with studies testing antigen-based therapies utilizing specific autoantibody positivity criteria. The DPT-1 oral and parental insulin studies and TrialNet oral insulin study identified participants based on insulin autoantibody positivity and first-phase insulin response on intravenous (IV) glucose tolerance testing[17,24,28]. The TrialNet teplizumab prevention study only enrolled individuals with multiple autoantibody positivity and dysglycemia on oral glucose tolerance testing. Finally, a study testing glutamic acid decarboxylase (GAD) antigen therapy was limited to individuals who were GAD autoantibody positive[19]. Most prevention trials (9/11; 81%) were multicenter studies; 9/11 (82%) were also double-masked, while 2/11 (18%) had no masking. In addition to these 11 papers, two follow-up papers and two papers focused solely on precision analysis of treatment response were also identified (for a total of 15 papers). Overall, only two prevention studies reported a positive impact on time to islet autoantibody positivity or time to diabetes: the primary prevention study testing whey-based hydrolyzed vs. cow's milk formula[30] and the secondary prevention study testing teplizumab[21] (forest plot showing hazard ratios for all prevention studies in Fig. 2).

The 60 manuscripts generated from 45 trials in the new-onset T1D population included 42 primary trial papers, 6 trial longitudinal follow-up papers and 12 papers focused solely on

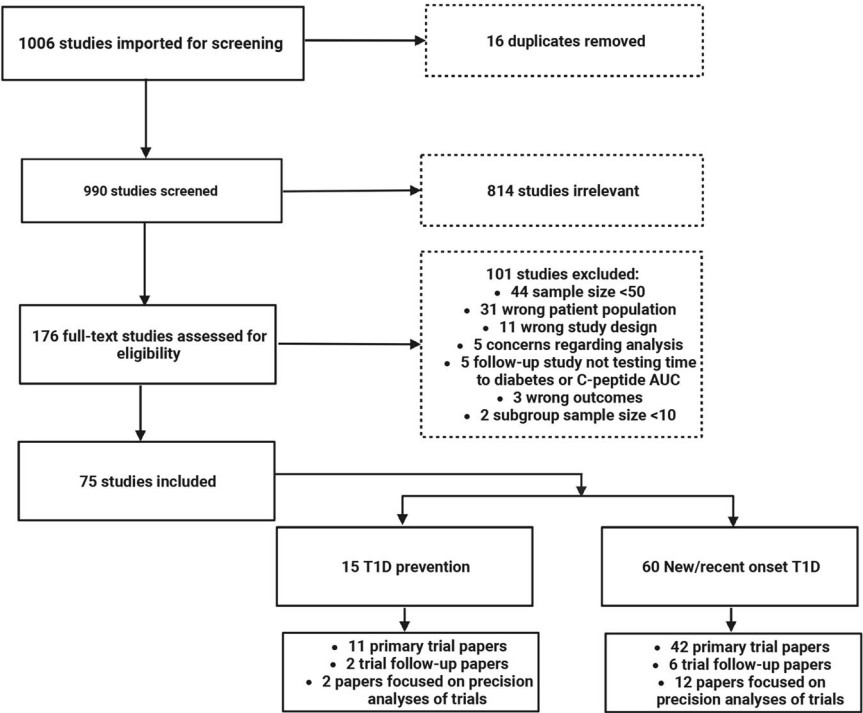

**Fig. 1 PRISMA flow diagram.** Flowchart displaying studies screened and excluded as part of abstract screening, then via full text review/eligibility assessment. 75 total papers were included in the extraction. AUC area under the curve; T1D Type 1 diabetes. This image was generated using Biorender.

**Table 1 Primary prevention studies.**

| Trial acronym | Population | Intervention | Multi-center? | Blinding | Primary outcome | Follow-up duration | Positive? | Hazard ratio (95%CI) vs. control |
|---|---|---|---|---|---|---|---|---|
| Hummel 2011[22] (BABYDIET) | 150 infants with a first-degree family history of T1D and high-risk HLA genotypes | Late (12 months) vs. early (6 months) gluten exposure | No | None | Aabs | 3 years (range 3.0–10.0) | No | 1.3 (0.6,3.0) |
| Vaarala 2012[30] (FINDIA) | 1113 infants with high-risk HLA genotypes | • Whey-based hydrolyzed vs. cow's milk formula | Yes | Double | Aabs | n/a | • No | • 0.82 (0.38–1.7) |
| | | • Insulin-free whey-based vs. cow's milk formula | | | | | • Yes | • 0.24 (0.08–0.72) |
| Knip 2018[23] (TRIGR) | 2159 infants with a first-degree family history of T1D and high-risk HLA genotypes | Extensively hydrolyzed casein formula vs. conventional formula | Yes | Double | Time to diabetes | 11.5 years (Q1–Q3, 10.2–12.8 | No | 1.1 (0.8 to 1.5) |

*T1D type 1 diabetes, HLA human leukocyte antigen, Aab autoantibody, FINDIA Finnish Dietary Intervention Trial for the Prevention of Type 1 Diabetes, n/a not applicable, TRIGR trial to reduce IDDM in the genetically at risk, Aab+ autoantibody positive.*

precision analyses of treatment response (Fig. 1). Additional characteristics of these 60 papers are summarized in Supplementary Table 1. Here, except for variable age criteria, inclusion criteria were more homogeneous than in prevention studies, typically including participants with a clinical diagnosis of T1D (usually with islet autoantibody positivity) and C-peptide above a certain cutoff. Of the 43 trials, 30 (70%) included both adults and children, 9 (21%) tested only children, and 4 (9%) were performed solely in adults. Five trials had inclusion criteria that included positivity for a specific islet autoantibody. Trials described were typically multicenter studies (39/43; 91%) and double-masked (35/43; 81%). Two studies were single-masked, two described only masked outcomes testing, three had no masking, and masking was not described in one study.

A measure of beta cell function was by far the most common primary outcome specified amongst new-onset trials (single primary outcome in 33/43 (77%), co-primary outcome in 2/43; 5%), although other studies used HbA1c and/or insulin dose and one study used T1D remission. Primary outcome was not specified in 5 trials. All follow-up studies focused on a measure of beta cell function. Trials reporting a measure of beta cell function as the primary outcome most commonly utilized mean C-peptide AUC from a mixed meal tolerance test; values for these data were available for 32/35 primary trials and 5/6 follow-up studies and are summarized in Supplementary Table 2. Of trial manuscripts reporting these data, less than a fourth identified a positive effect of the intervention on mean C-peptide AUC. These included trials testing imatinib mesylate, low-dose anti-thymocyte globulin, teplizumab (anti-CD3 antibody), otelixizumab (anti-CD3 antibody), abatacept (CTLA4-Ig), rituximab (anti-CD20 antibody), golimumab (anti-TNF-alpha), recombinant IFN-alpha, and combination of anti-IL-21 antibody with liraglutide.

**Precision analyses focused on features associated with disease-modifying treatment response.** To determine whether there were specific individual features that impacted response to treatment (genetic, metabolic, immune), we assessed papers that included this type of precision analysis. Two papers from prevention and 12 papers from new onset studies focused solely on precision analyses of treatment response (i.e., no analysis of primary trial or longitudinal follow-up analysis of primary trial). An additional 43 papers also included some aspect of precision analysis (summarized in Supplementary Table 3). Of 57 total papers identified, most (38/57; 67%) were primary trial papers with a section focused on features of treatment response. Just over half (5/8) of the primary trial follow-up papers included precision analyses of treatment response; these represented only 8.8% of the 57 papers including a precision analysis.

While precision analysis of treatment response was commonly reported, this was rarely pre-specified, occurring in just 16/57 (28%) of papers studied (Fig. 3b). Prespecified precision analyses were more common in primary trial or primary trial follow-up papers. For primary trials, 34% (13/38) of precision analyses were prespecified, and 10.5% (4/38) had both pre-specified and post hoc analyses. For follow-up papers, 40% (2/5) were pre-specified. In contrast, only 7% (1/14) of papers focused specifically on precision analyses described a prespecified analysis plan. Analyses tended to identify a positive relationship with treatment effect (Fig. 3c), with 37/57 (67%) studies identifying a significant relationship between a feature and treatment response. This was more prevalent for precision analyses in primary trial follow-up papers (5/5; 100%) and in precision analysis-only papers (13/14; 93%).

Because sample sizes inevitably decrease as groups are subdivided for precision analyses, we next looked at sample sizes for the precision subgroups. Only slightly over half (30/57) of

**Table 2 Secondary prevention studies.**

| Trial acronym | Population | Intervention | Multi-center? | Blinding | Primary Outcome | Follow-up duration | Positive? | Hazard ratio (95%CI) vs. control |
|---|---|---|---|---|---|---|---|---|
| Näntö-Salonen 2008[26] | 264 infants with high-risk HLA genotype and their siblings with high-risk HLA and multiple Aab+ | Intranasal daily recombinant human short-acting insulin vs. placebo | Yes | Double | Time to diabetes | Insulin: 1.7 years (IQR 0.7–3.0) Placebo: 2.0 years (IQR 0.8–3.2) | No | • Infants: 1.2 (0.68–2.0) • Infants + siblings: 0.98 (0.67–1.4) |
| Lampeter 1998[25] (DENIS) | 55 Islet-cell Aab+ siblings of individuals with T1D | 1.2 g/m²/day Endur-Amide (nicotinamide) vs. placebo | Yes | Double | Time to diabetes | 2.1 years, maximum 3.8 | No | 0.79 (0.25–3.4) |
| Gale 2004[20] (ENDIT) | 552 Islet-cell Aab+ relatives with nondiabetic OGTT | 1.2 g/m² po modified release nicotinamide x 5 years vs. placebo | Yes | Double | Time to diabetes | 5 years (intended for all, but only reached by 88%) | No | 1.1 (0.78, 1.5) |
| Skyler 2002[17] (DPT-1) | 339 Islet-cell Aab+ first-degree relatives with the absence of low-risk HLA and low first-phase insulin response or dysglycemia | 0.25U/kg ultralente + annual 4-day continuous insulin infusion vs. no intervention | Yes | None | Time to diabetes | 1345 days, IQR 784–1737 | No | 0.96 (0.69–1.3) |
| Skyler 2005[28] (DPT-1) | 372 Islet-cell and insulin Aab + relatives with the absence of low-risk HLA, higher first-phase insulin response, and normal OGTT | Oral insulin (7.5 mg/day/day) vs. placebo | Yes | Double | Time to diabetes | 4.3 years (IQR: 928–1988 days) | No | 0.76 (0.51,1.1) |
| • Vehik 2011[31] (F/u) • Butty 2008[18] (Precision) | • 303/372 • 638 from parenteral and oral insulin trials | | | | | • 9.1 years • n/a | | |
| Krischer 2017[24] (TN07) • Sosenko 2020[29] (Precision: DPT-1 and TN07) | 560 Multiple Aab+ relatives with insulin Aab + and high or low first-phase insulin response • 208 with high DPTRS | 7.5 mg daily po recombinant human insulin vs. placebo | Yes | Double | Time to diabetes | 2.7 years (IQR 1.5–4.7 years) • n/a | No | 0.83 (0–1.07) • DPT-1: 0.494 (0.26, 0.96) TN07: 0.70 (0.43, 1.2) |
| EldingLarsson 2018[19] (DiAPREV-IT) | 50 Multiple Aab+ children with GAD Ab+ | 20 ug sc injections of GAD-Alum monthly x 2 vs. placebo | No | Double | Other: safety | 4.92 years (range: 0.47–5.0) | n/a | 0.77 (0.30, 1.9) |
| Herold 2019[21] (TN10) | 76 Multiple Aab+ relatives with dysglycemia | 14-day course of IV Teplizumab vs. placebo | Yes | Double | Time to diabetes | 745 days (range 74–2683) | Yes | 0.41(0.22–0.78) |
| • Sims 2021[27] (F/u) | | | | | | • 923 days | | • 0.457 |

Follow-up or precision studies describing a randomized trial that is already included in the table are listed as bulleted subheadings.
*T1D* type 1 diabetes, *HLA* human leukocyte antigen, *Aab* autoantigen, *Aab+* autoantibody-positive, *n/a* not applicable, *DENIS* The Dutch Nicotinamide Intervention Study, *ENDIT* European Nicotinamide Diabetes Intervention Trial, *OGTT* oral glucose tolerance test, *Po* per oral/orally, *DPT-1* Diabetes Prevention Trial Type 1 Diabetes, *F/u* follow-up, *TN07* TrialNet 07 trial, *DPTRS* diabetes prevention trial-type 1 risk score, *DiAPREV-IT* diabetes prevention-immune tolerance trial, *GAD* glutamic acid decarboxylase, *Sc* subcutaneous, *TN10* TrialNet 10 trial, *IV* intravenous.

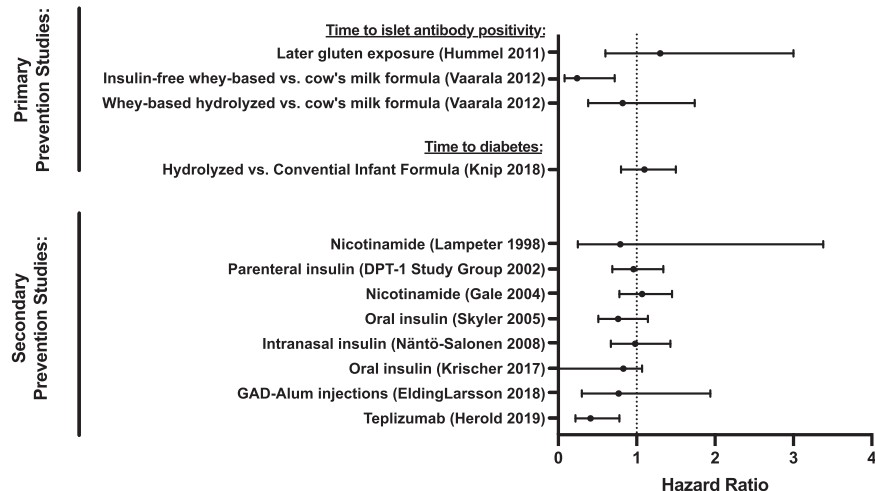

**Fig. 2 Relative effect of prevention therapies in individuals at risk for T1D.** Forest plot showing hazard ratio with 95% confidence intervals for primary prevention studies in genetically at-risk individuals and secondary prevention studies in individuals with elevated islet autoantibody titers. Primary prevention studies are divided by outcome—either time to islet autoantibody positivity or time to diabetes. All secondary prevention studies used time to diabetes as a primary outcome. DPT-1 Diabetes Prevention Trial Type 1 ; GAD glutamic acid decarboxylase.

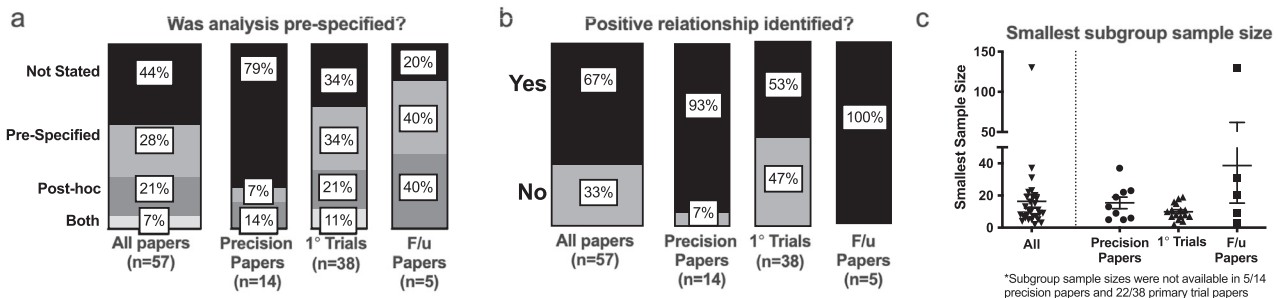

**Fig. 3 Precision analyses focused on treatment response were mostly part of primary trial papers, tended to be post hoc, and were biased toward positive findings. a** Stacked bar graphs showing relative frequencies and percentages of papers with precision analyses that were defined as prespecified, post hoc, or included both prespecified and post hoc analyses in the manuscript text. **b** Stacked bar graph displaying relative frequencies and percentages of papers reporting positive findings related to associations with treatment effects. **c** For papers that listed sample sizes of subgroups tested for differential treatment effects (only 53% of all papers with precision analyses), the smallest sample size reported is displayed, with mean and SEM indicated. F/u follow-up; n = 9 for precision papers; n = 16 for primary trials; n = 5 for f/u papers.

papers reported sample sizes for all subgroups defined by precision features. Within these 30 manuscripts, we observed wide variability in the sample sizes of the subgroups analyzed. Figure 3d displays reported values for the smallest subgroup sample size described. Overall median values were 11 (interquartile range of 7-19) participants, ranging from 2 to 128 participants.

Figure 4a displays the number of precision features tested for each paper. For all papers, the median number of features tested was 3 (interquartile range of 1–7). This tended to be higher in papers focused solely on precision analyses (median of 6.5 with several papers testing numerous subgroups as part of sequencing, array, or flow cytometry analysis). Forty-one papers analyzed multiple precision features. Of these applicable analyses, corrections for multiple comparisons were either not mentioned or not performed in 35/41 (85%) of papers, particularly for trials (100% of applicable papers with multiple comparisons not described or not performed) (Fig. 4b).

We next examined the types of features tested for relationships with treatment response (Fig. 4c). In trial papers and follow-up papers, age was most commonly tested (>75% of analyses), followed by a measure of beta cell function (>50% of analyses). Only 9/36 (25%) studies testing age identified a significant

relationship with treatment response; these were all in the new onset period[32,41,46,48,54,58,61,84,87]. Here, younger age groups showed improved treatment responses to teplizumab, ChAglyCD3, and Vitamin E. In contrast, older age was linked to a beneficial treatment response vs. placebo with high-dose antithymocyte globulin (ATG) and oral insulin (both studies with negative findings overall)[46,48]. One study showed that younger age was linked to a more rapid decline of C-peptide compared to placebo in Bacillus Calmette-Guerin (BCG) vaccine-treated individuals[32]. Baseline measures of beta cell function were linked to differences in treatment response in 10/26 (38%) of analyses where this relationship was tested[21,24,40,47,54,60,61,73,88,89]. In two papers focused on prevention studies, measures linked to worsened beta cell function were associated with an improved response to treatment (with oral insulin or teplizumab)[21,24]. Analyses testing trials in the new-onset period had split results: teplizumab, ChAglyCD3, linomide, and atorvastatin performed better compared to placebo in groups with better baseline beta cell function measures[40,54,60,61,88]. In contrast, canakinumab, imatinib mesylate, and the anti-IL-21/liraglutide combination showed stronger treatment effects in individuals with lower baseline beta cell function measures[47,73,89]. Taken in aggregate these results highlight evidence that baseline beta cell function may impact treatment

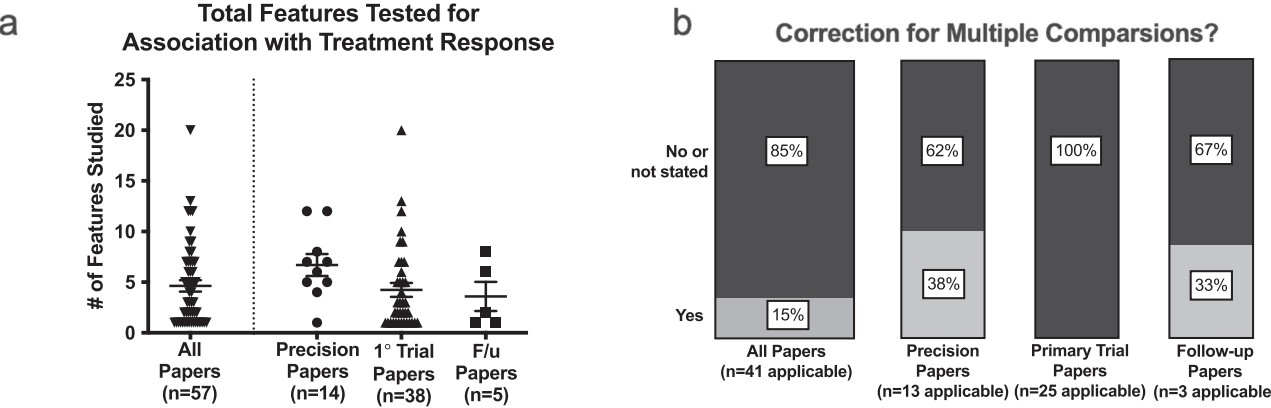

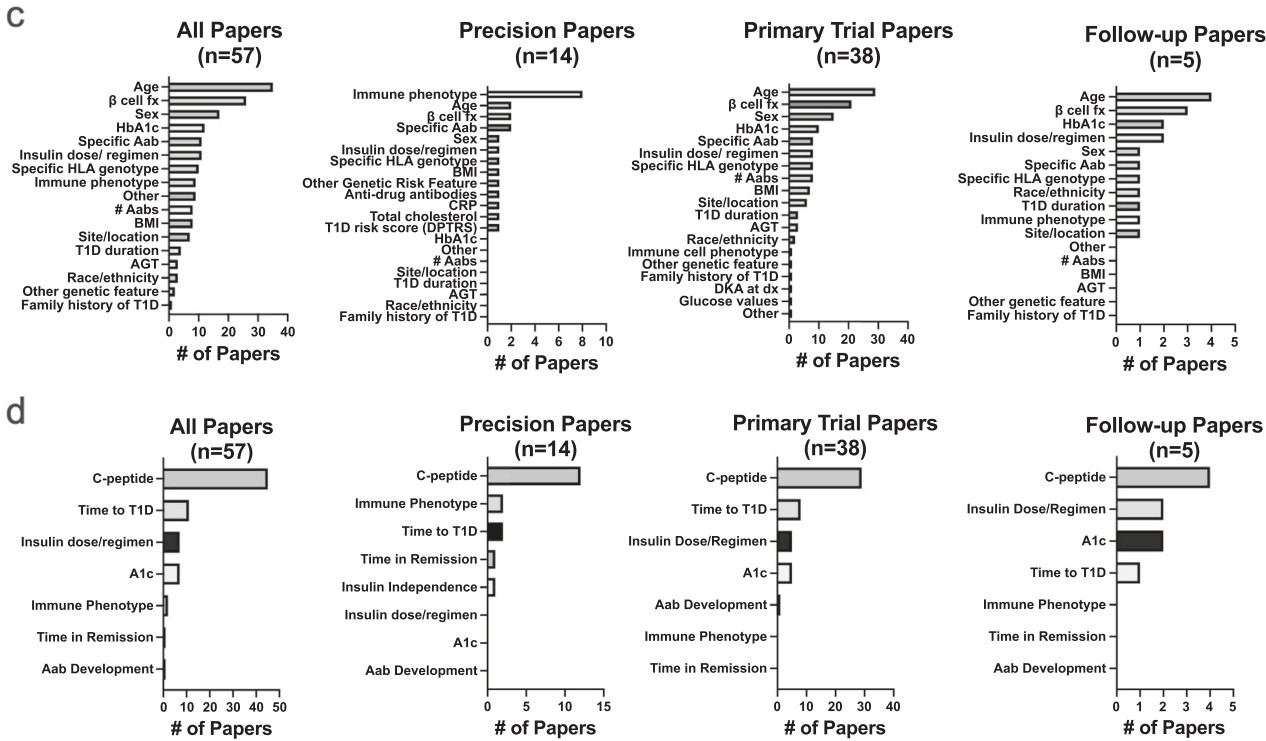

**Fig. 4 Precision analyses tested many features, most commonly age and beta cell function, infrequently corrected for multiple comparisons, and typically tested for differential impacts on a C-peptide-based measure. a** Total number of features tested for association with each treatment response, with mean and SEM indicated, for all papers with precision analyses. **b** Stacked bar graph showing relative frequencies and percentages of papers that did or did not correct for multiple comparisons. **c** Frequencies of individual features tested for associations with treatment response. **d** Frequencies of outcomes utilized to assess for the presence of any features associated with differential treatment response. The C-peptide measure category was inclusive of any measure of beta cell function, including mixed meal area under the curve, stimulated C-peptide values, fasting C-peptide values, etc. F/u follow-up, fx function, Hba1c hemoglobin A1c, Aab autoantibody, HLA human leukocyte antigen, BMI body mass index, T1D type 1 diabetes, AGT abnormal glucose tolerance, CRP C-reactive Protein, DPTRS diabetes prevention trial-type 1 risk score, DKA diabetes ketoacidosis, Dx diagnosis.

response, but the direction of impact likely varies by treatment used and stage of disease.

Interestingly, in contrast to primary trial papers, precision papers most commonly tested relationships of an immune cell phenotype with treatment response (57%). Because only two papers identified included a favorable response to time to type 1 diabetes diagnosis, treatment response was assessed using a range of alternative outcomes (Fig. 4d). For all types of papers, a measure of C-peptide was most commonly used as an alternative outcome to gauge treatment response (range of 44-68%).

**Risk of bias/quality assessments**. A finding impacting studies in all categories was a lack of racial and ethnic diversity in participant populations. Data on participant race were available in less than a third (23/75) of total papers; for reported papers, participants self-reporting as white race comprised a median of 92% of the total study population (interquartile range of 88-96%). Ethnicity was reported in 20 papers; within these manuscripts, participants self-reporting as identifying with a Hispanic ethnicity comprised a median of 5% of study participants (interquartile range of 3–9%).

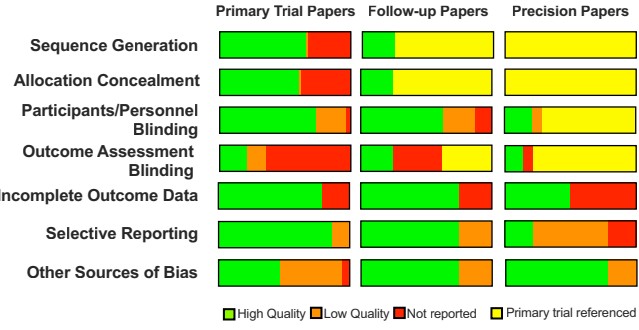

**Fig. 5 Risk of bias assessments for each paper category. Bias was assessed using Covidence's Cochrane risk of bias tool.** For sequence generation, allocation concealment, and blinding categories, raters had the option of selecting high quality (green), low quality (orange), not reported (red), or that a decision could not be made because of primary trial was referenced in methods (yellow). For incomplete outcome data, raters only had the option to choose high quality/data provided (green) or low quality/ data not provided (red). For selective reporting, raters had the option to select high-quality/primary endpoint predefined (green), low-quality/ primary endpoint not defined (orange), or low-quality/not reported (red). For other sources of bias, raters had the option to select high quality/none (green), low quality/bias present but identified and considered (orange), or low quality/obvious bias present and not addressed (red). Data are shown as absolute frequencies.

When assessing additional risks of bias, we found that many papers did not include details sufficient to assess these risks (Fig. 5). Although over half of primary trial papers were considered to utilize high-quality methods for sequence generation and allocation concealment, 32–37% did not describe methods adequately for assessment. Follow-up and precision papers infrequently described these methods, commonly citing a primary trial paper instead (75–100%). Blinding was described more frequently, with at least double blinding in 63–74% of follow-up and primary trial papers, although 23–25% had single or no blinding. In contrast, blinding of outcome assessments was either not described or did not occur in 79% of primary trial papers. Most precision papers referenced primary papers and so blinding was challenging to assess. Completeness of outcome data reporting was assessed by considering reasons and numbers for attrition or exclusion in studies. Reporting of outcomes was overall high quality for trials and follow-up studies (75–79%). This was less frequently the case for precision papers, only half of which reported on reasons for incomplete outcome data. While the large majority (87%) of trial papers described a prespecified primary endpoint, only 75% of follow-up papers and 21% of precision papers solely included analyses that were noted to be prespecified. Additional sources of bias were identified in 33/75 total papers (44%), and these biases were also acknowledged by study authors. These were most frequently acknowledged funding or support by a pharmaceutical company. However, another source of bias that was not addressed as a limitation by the authors was identified in 3 papers (all primary trial papers). No concerns for other unacknowledged sources of bias were identified in follow-up studies and precision studies.

## Discussion

We filled a gap in the T1D literature by systematically reviewing 25 years of large randomized controlled trials focused on T1D disease-modifying therapies, as well as precision analyses focused on identifying features of treatment response. Several themes in the literature were identified. Immunotherapies were the most common disease-modifying agents tested, and a resounding

majority of these agents were tested in "new-onset" trials after a clinical diagnosis of T1D. Of the 17 interventions that showed benefit in slowing T1D progression or preserving endogenous insulin secretion, only two were tested prior to clinical disease onset. Primary trial outcome papers most commonly included precision analyses testing the impacts of baseline age and beta cell function on treatment response, while post hoc precision analysis papers primarily focused on immune phenotypes.

Based on clinical heterogeneity observed in phenotypes of T1D progression and severity, a precision-based concept that has gained popularity is the idea of the T1D "endotype", a T1D subtype "defined by a distinct functional or pathobiological mechanism (that is also tractable therapeutically)"[14]. Along these lines, trials designed to limit participant heterogeneity based on features associated with treatment response could theoretically allow for clearer determinations of effect and a greater number of positive trials. While trials were overall of high quality, a key take-home message is that the current review did not identify high-quality clinical trial data supporting the existence of individual features consistently linked to therapeutic response and justifying translation to clinical care. Below, we highlight important considerations identified by this analysis for the future applicability of precision medicine to T1D disease modification.

**Standardization of approaches to outcomes for precision analyses.** Time to T1D was the most consistent primary outcome of T1D prevention studies, while the vast majority of new-onset studies used mixed meal-stimulated C-peptide AUC, consistent with the consensus recommendations by Palmer and colleagues[92]. However, precision endpoints were much more variable and would benefit from a similar consensus definition of "responders" to disease-modifying agents within larger trial populations. Strategies applied have included time to diabetes, insulin use, stratification based on changes in C-peptide, and identification of individuals exhibiting less C-peptide loss compared to placebo[21,29,61,67]. Although C-peptide was by far the most frequent outcome measure used to identify differential treatment responses, approaches to stratify based on C-peptide were highly variable. Consistent approaches, such as a quantifiable metric based on expected values[93] will allow better comparison of features associated with treatment response across trials.

**A recurring role for age and measures of beta cell function.** Age and measures of beta cell function were most frequently identified as factors associated with differential treatment response in primary trial and primary trial follow-up papers. For example, younger age was linked to improved treatment response in several new-onset trials using CD3-based agents[54,58,61,87]. The association of age with treatment response is in keeping with the strong associations of age to features of T1D in many observational and natural history studies, before and after clinical diagnosis[14,94–96]. Differences in pancreas histology have been identified in donors with younger age of diagnosis[97,98]. However, it is unclear whether differences in treatment response linked to age are associated with differences in underlying disease pathophysiology vs. differences in severity or progression of T1D at the time of treatment. The observation that age differentially impacts outcomes in different trials, in addition to stratification of both immune phenotypes and beta cell function by age, supports the idea that the underlying biological reasons for age associations could be linked to mechanisms and are important to consider in future trial designs and potentially in future precision therapy.

Thirty-eight percent of studies testing the impacts of baseline beta cell function showed a significant link to treatment response,

consistent with the substantial body of literature identifying an ongoing dialog between autoimmunity and the beta cell in T1D[94,99–106]. Interestingly, findings somewhat differed depending on the stage of intervention. Here, two unique prevention studies testing oral insulin and teplizumab showed that worse beta cell function was associated with improved treatment outcomes compared to placebo[21,24]. In contrast, CD3-based therapy trials after disease onset showed an association between higher baseline insulin secretion and improved outcomes[54,60,61]. These differences highlight the importance of considering the disease stage in the design and interpretation of intervention efforts[107]. Especially at earlier stages in the disease process, abnormalities in beta cell function could allow insight into a therapeutic window during active disease or immune attack, and optimal timing of therapy[108]. In contrast, in more advanced diseases after diagnosis, associations with differences in beta cell function could reflect differences in the degree of disease progression, and so amenability to prolonged preservation of a larger residual beta cell mass. Differences in the relationships between beta cell function measures and outcomes for different agents in the new onset period also highlight agent mechanism of action as a critical consideration for designs incorporating beta cell function into the stratification of trial populations and precision approaches to disease-modifying therapy.

**Reproducible biomarkers linked to underlying disease pathology.** Specific autoantibodies and immune cell phenotypes were also linked to treatment response for multiple agents. An important consideration in these types of assays is reproducibility. The T1D field has been strengthened by an international standardization program for autoantibody measurement that underpinned the development of T1D staging criteria[109]. If novel mechanistic markers (immune, metabolic, or other) can be used to predict treatment response, then similar scrutiny and standardization of these markers will be needed for cross-study comparisons and successful implementation.

**A need for pre-specified, appropriately powered precision analyses.** Our review identified important methodologic considerations with many precision analyses. While there were multiple notable and interesting results, most trial manuscripts (primary or follow-up) included precision analyses that were not prespecified, which decreased the quality ranking of these studies. Corrections for multiple comparisons were rare. Additionally, subgroup sizes were infrequently reported, but when available, these group sizes were highly variable and as small as $n = 2$ participants. Papers also tended to show positive results, raising concern for publication bias.

While these issues are a known limitation of hypothesis-generating exploratory analyses, follow-up studies focusing on testing positive findings a priori will be critical to the application of clinically meaningful precision medicine. An example of the necessity of hypothesis testing was the TrialNet oral insulin prevention study, which was prospectively designed to test a responder subgroup identified in the Diabetes Prevention Trial Type 1 (DPT-1) trial with high insulin autoantibody titers, and ultimately found no significant impact of treatment within this group[24]. Interestingly, within this trial, a significant protective effect of oral insulin was identified as part of a prespecified precision analysis of individuals with lower first-phase insulin response. Testing in future studies will be needed to understand the reproducibility of this finding. Another example of the application of this approach moving forward is the DIAGNODE 3 study (NCT05018585), which did not meet inclusion criteria for the current review but will prospectively test for a preferential benefit of GAD-alum injections in the HLA DR3-DQ2 population that was identified in post hoc analyses[71,110]. Based on frequent testing and existing studies suggesting impacts of baseline age and beta cell function as potential features associated with treatment response, pre-specified analyses for appropriately powered studies testing the impact of these precision features should be considered in trials moving forward.

**Impacts of the T1D staging system.** Time to T1D was the most consistent primary outcome of T1D prevention studies, but inclusion criteria for these studies varied widely across trials, including combinations of genetic risk, presence of islet autoantibodies, and changes in glycemia and/or beta cell function. Recent progress in understanding the natural history of T1D, particularly the high lifetime risk associated with progression from multiple autoantibodies to clinical T1D[111], led to a revision of the definition of T1D to include early stages of disease[9,112]. Stage 1 and stage 2 T1D are now defined by the presence of multiple autoantibodies without or with dysglycemia, whereas clinical disease is now considered as stage 3 T1D[9,113]. Because these stages were developed concurrently with many of the trials included in this review, these definitions were not applicable at the time of many of these trials, limiting our ability to apply staging categories to this review. As noted above, the timing of T1D stages impacts study feasibility (rate of progression, participant availability) and may be critical to intervention efficacy, highlighting the importance of considering the disease stage in the design and interpretation of intervention efforts. Moving forward, widespread adoption of the T1D staging system combined with increased screening efforts spurred by the recent positive teplizumab trial in stage 2 T1D may allow for increased numbers of trials in earlier stages of the disease.

**Limitations.** This study has several limitations. The heterogeneity of included papers limited our ability to perform metanalysis. For feasibility, we restricted our review of primary trials to those enrolling a minimum of 50 total participants. Because of this, some trials were not reviewed, including positive trials testing alefacept[114,115] and verapamil[116]. A large pediatric follow-up trial testing verapamil (positive outcome) and tight metabolic control with a hybrid closed loop (negative outcome) was published after the conclusion of our systematic review[117,118]. In addition, most studies reviewed did not report data on race or ethnicity. For those that did report these data, populations studied largely identified as non-Hispanic white. Barriers to screening of traditionally underrepresented populations is a recognized issue amongst T1D natural history and intervention studies[93,119]. This is especially important to address moving forward given the rising incidence of T1D in these populations[120].

In summary, our review identified noteworthy progress towards defining effective disease-modifying therapies for T1D but a need for better quality data to support the existence of individual features consistently linked to differences in treatment response. Our findings specifically highlight the need for standardization of precision outcome measures, reproducible biomarkers of disease pathology, and prespecified, adequately powered precision analyses. Reports of future trials would benefit from including adequate details to assess potential risks of bias.

## Data availability

All studies reviewed were identified via publicly available databases (PubMed and Embase). All included articles are outlined in Supplementary Information and Supplementary Data 1. Source data for the figures are included in Supplementary Data 2. Article review data supporting the findings of this study are available upon reasonable request from the corresponding author.

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

## Acknowledgements

We thank Krister Aronsson and Maria Bjorklund from Lund University for assistance with database searches and Russell de Souza from McMaster University for advice on critical appraisal. The ADA/EASD Precision Diabetes Medicine Initiative, within which this work was conducted, has received the following support: The Covidence license was funded by Lund University (Sweden) for which technical support was provided by Maria Björklund and Krister Aronsson (Faculty of Medicine Library, Lund University, Sweden). Administrative support was provided by Lund University (Malmö, Sweden), the University of Chicago (IL, USA), and the American Diabetes Association (Washington D.C., USA). The Novo Nordisk Foundation (Hellerup, Denmark) provided grant support for in-person writing group meetings (P.I.: L Phillipson, University of Chicago, IL). J.F.: DiabDocs K12 program 1K12DK133995-01 (DiMeglio, Maahs PIs), The Leona M. & Harry B. Helmsley Charitable-Trust Grant #2307-06126 (Felton PI). KG: The Leona M. and Harry B. Helmsley Charitable Trust and Sanford Health. R.A.O.: RAO had a UK MRC confidence in concept award to develop a type 1 diabetes GRS biochip with Randox R&D and has ongoing research funding from Randox; and has research funding from a Diabetes UK Harry Keen Fellowship (16/0005529), National Institute of Diabetes and Digestive and Kidney Diseases grants (NIH R01 DK121843–01 and U01DK127382–01), JDRF (3-SRA-2019–827-S-B, 2-SRA-2022–1261-S-B, 2-SRA-2002–1259-S-B, 3-SRA-2022–1241-S-B, and 2-SRA-2022–1258-M-B), and The Larry M and Leona B Helmsley Charitable Trust; and is supported by the National Institute for Health and Care Research Exeter Biomedical Research Centre. The views expressed are those of the author(s) and not necessarily those of the National Institutes for Health Research or the Department of Health and Social Care. L.A.D.: NIH for Trial-Net U01DK106993/6163-1082-00-BO, DiabDocs K12 program 1K12DK133995-01, CTSI UL1TR001108-01, C.E.M.: R01DK093954, R01DK127236, U01DK127786, R01DK127308, and UC4DK104166, U54DK118638, P30 P30 DK097512), a US Department of Veterans Affairs Merit Award (I01BX001733), grants from the JDRF (3-IND-2022-1235-I-X) and Helmsley Charitable Trust (2207-05392), and gifts from the Sigma Beta Sorority, the Ball Brothers Foundation, and the George and Frances Ball Foundation. HI: K23DK129799; RJ: NIH R03-DK127472 and The Leona M. and Harry B. Helmsley Charitable Trust (2103-05094); S.A.L.: NIH NIAID R01 AI141952 (PI), NIH NCI R01 CA231226 (Other support), NIH NIAID 1 R01HL149676 (Other support), NIH NIDDK 1UC4DK117483 (subaward), JDRF 3-SRA-2019-851-M-B; S.O.G.: NIH R01 DK121843–01; S.R.: R01 DK122586, THE LEONA M AND HARRY B HELMSLEY CHARITABLE TRUST 2204-05134; J.W.: JDRF 2-SRA-2022-1282-M-X, 3-SRA-2022-1095-M-B, 4-SRA-2022-1246-M-N, 3-SRA-2023-1374-M-N.; M.R.: NIH NIDDK R01DK124395 and R01DK121843; R01DK121929A1, R01DK133881, U01DK127786, U01 DK127382 (E.K.S.). Effort from this grant (to E.K.S., H.I., J.F.) is also supported by Grant 2021258 from the Doris Duke Charitable Foundation through the COVID-19 Fund to Retain Clinical Scientists collaborative grant program and was made possible through the support of Grant 62288 from the John Templeton Foundation. No funders played any role in the design, implementation, or writing of this review.

## Author contributions

J.L.F., K.J.G., R.A.O., M.J.R., and E.K.S. designed the project, performed a systematic review, interpreted results, and wrote and edited the manuscript. C.S., S.A.L., S.O.G., S.S.R., G.S.F.M., C.E.M., L.A.D., H.M.I., A.K.S., D.D., R.K.J., M.U., S.G., and J.M.W. contributed to the design of the project, performed a systematic review, and edited the manuscript.

## Competing interests

E.K.S. has received compensation for educational lectures from Medscape, ADA, and MJH Life Sciences and as a consultant for DRI Healthcare. C.E.M. reported serving on advisory boards for Provention Bio, Isla Technologies, MaiCell Technologies, Avotres, DiogenyX, and Neurodon; receiving in-kind research support from Bristol Myers Squibb and Nimbus Pharmaceuticals; and receiving investigator-initiated grants from Lilly Pharmaceuticals and Astellas Pharmaceuticals. L.A.D. reports research support to institutions from Dompe, Lilly, Mannkind, Prevention, Zealand, and consulting relationships with Abata and Vertex. R.A.O. had a UK MRC Confidence in concept grant to develop a T1D GRS biochip with Randox Ltd and has ongoing research funding from Randox R & D. No other authors report any relevant conflicts of interest.

## Additional information

[1]Department of Pediatrics, Center for Diabetes and Metabolic Diseases, Indianapolis, IN, USA. [2]Herman B Wells Center for Pediatric Research, Indiana University School of Medicine, Indianapolis, IN, USA. [3]Department of Pediatrics, Sanford School of Medicine, University of South Dakota, Sioux Falls, SD, USA. [4]Sanford Research, Sioux Falls, SD, USA. [5]NIHR Exeter Biomedical Research Centre (BRC), Academic Kidney Unit, University of Exeter, Devon, UK. [6]Department of Clinical and Biomedical Sciences, University of Exeter Medical School, Exeter, Devon, UK. [7]Royal Devon University Healthcare NHS Foundation Trust, Exeter, Devon, UK. [8]Center for Interventional Immunology, Benaroya Research Institute, Seattle, WA, USA. [9]Center for Translational Immunology, Benaroya Research Institute, Seattle, WA, USA. [10]Center for Public Health Genomics, University of Virginia, Charlottesville, VA, USA. [11]Richard L. Roudebush VAMC, Indianapolis, IN, USA. [12]Barbara Davis Center for Diabetes, Aurora, CO, USA. [13]Lifecourse Epidemiology of Adiposity and Diabetes (LEAD) Center, Aurora, CO, USA. [14]Department of Biomedical Informatics, University of Colorado Anschutz Medical Campus, Aurora, CO, USA. [15]Department of Epidemiology, Colorado School of Public Health, Aurora, CO, USA. [16]Department of Pediatrics, Baylor College of Medicine, Houston, TX, USA. [17]Department of Pediatrics, Diabetes Center; University of California at San Francisco, San Francisco, CA, USA. [18]Royal Melbourne Hospital Department of Diabetes and Endocrinology, Walter and Eliza Hall Institute, Parkville, VIC, Australia. [19]University of Melbourne Department of Medicine, Parkville, VIC, Australia. [20]Division of Pediatric Diabetes and Endocrinology, Texas Children's Hospital, Houston, TX, USA. [203]These authors jointly supervised this work: Maria J. Redondo, Emily K. Sims. *A list of authors and their affiliations appears at the end of the paper. ✉email: eksims@iu.edu

## ADA/EASD PMDI

Deirdre K. Tobias[21,22], Jordi Merino[23,24,25], Abrar Ahmad[26], Catherine Aiken[27,28], Jamie L. Benham[29], Dhanasekaran Bodhini[30], Amy L. Clark[31], Kevin Colclough[6], Rosa Corcoy[32,33,34], Sara J. Cromer[24,35,36], Daisy Duan[37], Jamie L. Felton[2,38,39], Ellen C. Francis[40], Pieter Gillard[41], Véronique Gingras[42,43], Romy Gaillard[44], Eram Haider[45], Alice Hughes[6], Jennifer M. Ikle[46,47], Laura M. Jacobsen[48], Anna R. Kahkoska[49], Jarno L. T. Kettunen[50,51,52], Raymond J. Kreienkamp[24,25,35,53], Lee-Ling Lim[54,55,56], Jonna M. E. Männistö[57,58], Robert Massey[45], Niamh-Maire Mclennan[59], Rachel G. Miller[60], Mario Luca Morieri[61,62], Jasper Most[63], Rochelle N. Naylor[64], Bige Ozkan[65,66], Kashyap Amratlal Patel[6], Scott J. Pilla[67,68], Katsiaryna Prystupa[69,70], Sridharan Raghavan[71,72], Mary R. Rooney[65,73], Martin Schön[69,70,74], Zhila Semnani-Azad[22], Magdalena Sevilla-Gonzalez[35,36,75], Pernille Svalastoga[76,77], Wubet Worku Takele[78], Claudia Ha-ting Tam[56,79,80], Anne Cathrine B. Thuesen[23], Mustafa Tosur[16,20,81], Amelia S. Wallace[65,73], Caroline C. Wang[73], Jessie J. Wong[82], Jennifer M. Yamamoto[83], Katherine Young[6], Chloé Amouyal[84,85], Mette K. Andersen[23], Maxine P. Bonham[86], Mingling Chen[87], Feifei Cheng[88], Tinashe Chikowore[36,89,90,91], Sian C. Chivers[92], Christoffer Clemmensen[23], Dana Dabelea[93], Adem Y. Dawed[45], Aaron J. Deutsch[25,35,36], Laura T. Dickens[94], Linda A. DiMeglio[2,38,39,95], Monika Dudenhöffer-Pfeifer[26], Carmella Evans-Molina[2,11,38,39], María Mercè Fernández-Balsells[96,97], Hugo Fitipaldi[26], Stephanie L. Fitzpatrick[98], Stephen E. Gitelman[99], Mark O. Goodarzi[100,101], Jessica A. Grieger[102,103], Marta Guasch-Ferré[22,104], Nahal Habibi[102,103], Torben Hansen[23], Chuiguo Huang[56,79], Arianna Harris-Kawano[2,38,39], Heba M. Ismail[2,38,39], Benjamin Hoag[105,106], Randi K. Johnson[14,15], Angus G. Jones[6,107], Robert W. Koivula[108], Aaron Leong[24,36,109], Gloria K. W. Leung[86], Ingrid M. Libman[110], Kai Liu[102], S. Alice Long[9], William L. Lowe Jr.[111], Robert W. Morton[112,113,114], Ayesha A. Motala[115], Suna Onengut-Gumuscu[116], James S. Pankow[117], Maleesa Pathirana[102,103], Sofia Pazmino[118], Dianna Perez[2,38,39], John R. Petrie[119], Camille E. Powe[24,35,36,120], Alejandra Quinteros[102], Rashmi Jain[3,121], Debashree Ray[73,122], Mathias Ried-Larsen[123,124], Zeb Saeed[125], Vanessa Santhakumar[21], Sarah Kanbour[67,126], Sudipa Sarkar[67], Gabriela S. F. Monaco[2,38,39], Denise M. Scholtens[127], Elizabeth Selvin[65,73], Wayne Huey-Herng Sheu[128,129,130], Cate Speake[8], Maggie A. Stanislawski[14], Nele Steenackers[118], Andrea K. Steck[131], Norbert Stefan[70,132,133], Julie Støy[134], Rachael Taylor[135], Sok Cin Tye[136,137], Gebresilasea Gendisha Ukke[78], Marzhan Urazbayeva[20,138], Bart Van der Schueren[118,139], Camille Vatier[140,141], John M. Wentworth[19,142,143], Wesley Hannah[144,145], Sara L. White[92,146], Gechang Yu[56,79], Yingchai Zhang[56,79], Shao J. Zhou[103,147], Jacques Beltrand[148,149], Michel Polak[148,149], Ingvild Aukrust[76,150], Elisa de Franco[6], Sarah E. Flanagan[6], Kristin A. Maloney[151], Andrew McGovern[6], Janne Molnes[76,150], Mariam Nakabuye[23], Pål Rasmus Njølstad[76,77], Hugo Pomares-Millan[26,152], Michele Provenzano[153], Cécile Saint-Martin[154], Cuilin Zhang[155,156], Yeyi Zhu[157,158], Sungyoung Auh[159], Russell de Souza[113,160], Andrea J. Fawcett[161,162], Chandra Gruber[163], Eskedar Getie Mekonnen[164,165], Emily Mixter[166], Diana Sherifali[113,167], Robert H. Eckel[168], John J. Nolan[169,170], Louis H. Philipson[166],

Rebecca J. Brown[159], Liana K. Billings[171,172], Kristen Boyle[93], Tina Costacou[60], John M. Dennis[6], Jose C. Florez[24,25,35,36], Anna L. Gloyn[46,47,173], Maria F. Gomez[26,174], Peter A. Gottlieb[131], Siri Atma W. Greeley[175], Kurt Griffin[3,4], Andrew T. Hattersley[6,107], Irl B. Hirsch[176], Marie-France Hivert[24,177,178], Korey K. Hood[82], Jami L. Josefson[161], Soo Heon Kwak[179], Lori M. Laffel[180], Siew S. Lim[78], Ruth J. F. Loos[23,181], Ronald C. W. Ma[56,79,80], Chantal Mathieu[41], Nestoras Mathioudakis[67], James B. Meigs[36,109,182], Shivani Misra[183,184], Viswanathan Mohan[185], Rinki Murphy[186,187,188], Richard Oram[6,107], Katharine R. Owen[108,189], Susan E. Ozanne[190], Ewan R. Pearson[45], Wei Perng[93], Toni I. Pollin[151,191], Rodica Pop-Busui[192], Richard E. Pratley[193], Leanne M. Redman[194], Maria J. Redondo[16,20,203], Rebecca M. Reynolds[59], Robert K. Semple[59,195], Jennifer L. Sherr[196], Emily K. Sims[2,38,39], Arianne Sweeting[197,198], Tiinamaija Tuomi[52,50,143], Miriam S. Udler[24,25,35,36], Kimberly K. Vesco[199], Tina Vilsbøll[200,201], Robert Wagner[69,70,202], Stephen S. Rich[116] & Paul W. Franks[22,26,108,114]

[21]Division of Preventative Medicine, Department of Medicine, Brigham and Women's Hospital and Harvard Medical School, Boston, MA, USA. [22]Department of Nutrition, Harvard T.H. Chan School of Public Health, Boston, MA, USA. [23]Novo Nordisk Foundation Center for Basic Metabolic Research, Faculty of Health and Medical Sciences, University of Copenhagen, Copenhagen, Denmark. [24]Diabetes Unit, Endocrine Division, Massachusetts General Hospital, Boston, MA, USA. [25]Center for Genomic Medicine, Massachusetts General Hospital, Boston, MA, USA. [26]Department of Clinical Sciences, Lund University Diabetes Centre, Lund University, Malmö, Sweden. [27]Department of Obstetrics and Gynaecology, the Rosie Hospital, Cambridge, UK. [28]NIHR Cambridge Biomedical Research Centre, University of Cambridge, Cambridge, UK. [29]Departments of Medicine and Community Health Sciences, Cumming School of Medicine, University of Calgary, Calgary, AB, Canada. [30]Department of Molecular Genetics, Madras Diabetes Research Foundation, Chennai, India. [31]Division of Pediatric Endocrinology, Department of Pediatrics, Saint Louis University School of Medicine, SSM Health Cardinal Glennon Children's Hospital, St. Louis, MO, USA. [32]CIBER-BBN, ISCIII, Madrid, Spain. [33]Institut d'Investigació Biomèdica Sant Pau (IIB SANT PAU), Barcelona, Spain. [34]Departament de Medicina, Universitat Autònoma de Barcelona, Bellaterra, Spain. [35]Programs in Metabolism and Medical & Population Genetics, Broad Institute, Cambridge, MA, USA. [36]Department of Medicine, Harvard Medical School, Boston, MA, USA. [37]Division of Endocrinology, Diabetes and Metabolism, Johns Hopkins University School of Medicine, Baltimore, MD, USA. [38]Department of Pediatrics, Indiana University School of Medicine, Indianapolis, IN, USA. [39]Center for Diabetes and Metabolic Diseases, Indiana University School of Medicine, Indianapolis, IN, USA. [40]Department of Biostatistics and Epidemiology, Rutgers School of Public Health, Newark, NJ, USA. [41]University Hospital Leuven, Leuven, Belgium. [42]Department of Nutrition, Université de Montréal, Montreal, QC, Canada. [43]Research Center, Sainte-Justine University Hospital Center, Montreal, QC, Canada. [44]Department of Pediatrics, Erasmus Medical Center, Rotterdam, The Netherlands. [45]Division of Population Health & Genomics, School of Medicine, University of Dundee, Dundee, UK. [46]Department of Pediatrics, Stanford School of Medicine, Stanford University, Stanford, CA, USA. [47]Stanford Diabetes Research Center, Stanford School of Medicine, Stanford University, Stanford, CA, USA. [48]University of Florida, Gainesville, FL, USA. [49]Department of Nutrition, University of North Carolina at Chapel Hill, Chapel Hill, NC, USA. [50]Helsinki University Hospital, Abdominal Centre/Endocrinology, Helsinki, Finland. [51]Folkhalsan Research Center, Helsinki, Finland. [52]Institute for Molecular Medicine Finland FIMM, University of Helsinki, Helsinki, Finland. [53]Department of Pediatrics, Division of Endocrinology, Boston Children's Hospital, Boston, MA, USA. [54]Department of Medicine, Faculty of Medicine, University of Malaya, Kuala Lumpur, Malaysia. [55]Asia Diabetes Foundation, Hong Kong SAR, China. [56]Department of Medicine & Therapeutics, Chinese University of Hong Kong, Hong Kong SAR, China. [57]Departments of Pediatrics and Clinical Genetics, Kuopio University Hospital, Kuopio, Finland. [58]Department of Medicine, University of Eastern Finland, Kuopio, Finland. [59]Centre for Cardiovascular Science, Queen's Medical Research Institute, University of Edinburgh, Edinburgh, UK. [60]Department of Epidemiology, University of Pittsburgh, Pittsburgh, PA, USA. [61]Metabolic Disease Unit, University Hospital of Padova, Padova, Italy. [62]Department of Medicine, University of Padova, Padova, Italy. [63]Department of Orthopedics, Zuyderland Medical Center, Sittard-Geleen, The Netherlands. [64]Departments of Pediatrics and Medicine, University of Chicago, Chicago, IL, USA. [65]Welch Center for Prevention, Epidemiology, and Clinical Research, Johns Hopkins Bloomberg School of Public Health, Baltimore, MD, USA. [66]Ciccarone Center for the Prevention of Cardiovascular Disease, Johns Hopkins School of Medicine, Baltimore, MD, USA. [67]Department of Medicine, Johns Hopkins University, Baltimore, MD, USA. [68]Department of Health Policy and Management, Johns Hopkins University Bloomberg School of Public Health, Baltimore, MD, USA. [69]Institute for Clinical Diabetology, German Diabetes Center, Leibniz Center for Diabetes Research at Heinrich Heine University Düsseldorf, Auf'm Hennekamp 65, 40225 Düsseldorf, Germany. [70]German Center for Diabetes Research (DZD), Ingolstädter Landstraße 1, 85764 Neuherberg, Germany. [71]Section of Academic Primary Care, US Department of Veterans Affairs Eastern Colorado Health Care System, Aurora, CO, USA. [72]Department of Medicine, University of Colorado School of Medicine, Aurora, CO, USA. [73]Department of Epidemiology, Johns Hopkins Bloomberg School of Public Health, Baltimore, MD, USA. [74]Institute of Experimental Endocrinology, Biomedical Research Center, Slovak Academy of Sciences, Bratislava, Slovakia. [75]Clinical and Translational Epidemiology Unit, Massachusetts General Hospital, Boston, MA, USA. [76]Mohn Center for Diabetes Precision Medicine, Department of Clinical Science, University of Bergen, Bergen, Norway. [77]Children and Youth Clinic, Haukeland University Hospital, Bergen, Norway. [78]Eastern Health Clinical School, Monash University, Melbourne, VIC, Australia. [79]Laboratory for Molecular Epidemiology in Diabetes, Li Ka Shing Institute of Health Sciences, The Chinese University of Hong Kong, Hong Kong, China. [80]Hong Kong Institute of Diabetes and Obesity, The Chinese University of Hong Kong, Hong Kong, China. [81]Children's Nutrition Research Center, USDA/ARS, Houston, TX, USA. [82]Stanford University School of Medicine, Stanford, CA, USA. [83]Internal Medicine, University of Manitoba, Winnipeg, MB, Canada. [84]Department of Diabetology, APHP, Paris, France. [85]Sorbonne Université, INSERM, NutriOmic team, Paris, France. [86]Department of Nutrition, Dietetics, and Food, Monash University, Melbourne, VIC, Australia. [87]Monash Centre for Health Research and Implementation, Monash University, Clayton, VIC, Australia. [88]Health Management Center, The Second Affiliated Hospital of Chongqing Medical University, Chongqing Medical University, Chongqing, China. [89]MRC/Wits Developmental Pathways for Health Research Unit, Department of Paediatrics, Faculty of Health Sciences, University of the Witwatersrand, Johannesburg, South Africa. [90]Channing Division of Network Medicine, Brigham and Women's Hospital, Boston, MA, USA. [91]Sydney Brenner Institute for Molecular Bioscience, Faculty of Health Sciences, University of the Witwatersrand, Johannesburg, South Africa. [92]Department of Women and Children's Health, King's College London, London, UK. [93]Lifecourse Epidemiology of Adiposity and Diabetes (LEAD) Center, University of Colorado Anschutz Medical Campus, Aurora, CO,

USA. [94]Section of Adult and Pediatric Endocrinology, Diabetes and Metabolism, Kovler Diabetes Center, University of Chicago, Chicago, USA. [95]Department of Pediatrics, Riley Hospital for Children, Indiana University School of Medicine, Indianapolis, IN, USA. [96]Biomedical Research Institute Girona, IdIBGi, Girona, Spain. [97]Diabetes, Endocrinology and Nutrition Unit Girona, University Hospital Dr Josep Trueta, Girona, Spain. [98]Institute of Health System Science, Feinstein Institutes for Medical Research, Northwell Health, Manhasset, NY, USA. [99]University of California at San Francisco, Department of Pediatrics, Diabetes Center, San Francisco, CA, USA. [100]Division of Endocrinology, Diabetes and Metabolism, Cedars-Sinai Medical Center, Los Angeles, CA, USA. [101]Department of Medicine, Cedars-Sinai Medical Center, Los Angeles, CA, USA. [102]Adelaide Medical School, Faculty of Health and Medical Sciences, The University of Adelaide, Adelaide, Australia. [103]Robinson Research Institute, The University of Adelaide, Adelaide, Australia. [104]Department of Public Health and Novo Nordisk Foundation Center for Basic Metabolic Research, Faculty of Health and Medical Sciences, University of Copenhagen, 1014 Copenhagen, Denmark. [105]Division of Endocrinology and Diabetes, Department of Pediatrics, Sanford Children's Hospital, Sioux Falls, SD, USA. [106]University of South Dakota School of Medicine, E Clark St, Vermillion, SD, USA. [107]Royal Devon University Healthcare NHS Foundation Trust, Exeter, UK. [108]Oxford Centre for Diabetes, Endocrinology and Metabolism, University of Oxford, Oxford, UK. [109]Division of General Internal Medicine, Massachusetts General Hospital, Boston, MA, USA. [110]UPMC Children's Hospital of Pittsburgh, Pittsburgh, PA, USA. [111]Department of Medicine, Northwestern University Feinberg School of Medicine, Chicago, IL, USA. [112]Department of Pathology & Molecular Medicine, McMaster University, Hamilton, ON, Canada. [113]Population Health Research Institute, Hamilton, ON, Canada. [114]Department of Translational Medicine, Medical Science, Novo Nordisk Foundation, Tuborg Havnevej 19, 2900 Hellerup, Denmark. [115]Department of Diabetes and Endocrinology, Nelson R Mandela School of Medicine, University of KwaZulu-Natal, Durban, South Africa. [116]Center for Public Health Genomics, Department of Public Health Sciences, University of Virginia, Charlottesville, VA, USA. [117]Division of Epidemiology and Community Health, School of Public Health, University of Minnesota, Minnesota, MN, USA. [118]Department of Chronic Diseases and Metabolism, Clinical and Experimental Endocrinology, KU Leuven, Leuven, Belgium. [119]School of Health and Wellbeing, College of Medical, Veterinary and Life Sciences, University of Glasgow, Glasgow, UK. [120]Department of Obstetrics, Gynecology, and Reproductive Biology, Massachusetts General Hospital and Harvard Medical School, Boston, MA, USA. [121]Sanford Children's Specialty Clinic, Sioux Falls, SD, USA. [122]Department of Biostatistics, Johns Hopkins Bloomberg School of Public Health, Baltimore, MD, USA. [123]Centre for Physical Activity Research, Rigshospitalet, Copenhagen, Denmark. [124]Institute for Sports and Clinical Biomechanics, University of Southern Denmark, Odense, Denmark. [125]Department of Medicine, Division of Endocrinology, Diabetes and Metabolism, Indiana University School of Medicine, Indianapolis, IN, USA. [126]AMAN Hospital, Doha, Qatar. [127]Department of Preventive Medicine, Division of Biostatistics, Northwestern University Feinberg School of Medicine, Chicago, IL, USA. [128]Institute of Molecular and Genomic Medicine, National Health Research Institutes, Taipei City, Taiwan. [129]Divsion of Endocrinology and Metabolism, Taichung Veterans General Hospital, Taichung, Taiwan. [130]Division of Endocrinology and Metabolism, Taipei Veterans General Hospital, Taipei, Taiwan. [131]Barbara Davis Center for Diabetes, University of Colorado Anschutz Medical Campus, Aurora, CO, USA. [132]University Hospital of Tübingen, Tübingen, Germany. [133]Institute of Diabetes Research and Metabolic Diseases (IDM), Helmholtz Center Munich, Neuherberg, Germany. [134]Steno Diabetes Center Aarhus, Aarhus University Hospital, Aarhus, Denmark. [135]University of Newcastle, Newcastle upon Tyne, UK. [136]Sections on Genetics and Epidemiology, Joslin Diabetes Center, Harvard Medical School, Boston, MA, USA. [137]Department of Clinical Pharmacy and Pharmacology, University Medical Center Groningen, Groningen, The Netherlands. [138]Gastroenterology, Baylor College of Medicine, Houston, TX, USA. [139]Department of Endocrinology, University Hospitals Leuven, Leuven, Belgium. [140]Sorbonne University, Inserm U938, Saint-Antoine Research Centre, Institute of Cardiometabolism and Nutrition, 75012 Paris, France. [141]Department of Endocrinology, Diabetology and Reproductive Endocrinology, Assistance Publique-Hôpitaux de Paris, Saint-Antoine University Hospital, National Reference Center for Rare Diseases of Insulin Secretion and Insulin Sensitivity (PRISIS), Paris, France. [142]Royal Melbourne Hospital Department of Diabetes and Endocrinology, Parkville, VIC, Australia. [143]Walter and Eliza Hall Institute, Parkville, VIC, Australia. [144]Deakin University, Melbourne, VIC, Australia. [145]Department of Epidemiology, Madras Diabetes Research Foundation, Chennai, India. [146]Department of Diabetes and Endocrinology, Guy's and St Thomas' Hospitals NHS Foundation Trust, London, UK. [147]School of Agriculture, Food and Wine, University of Adelaide, Adelaide, SA, Australia. [148]Institut Cochin, Inserm U 10116, Paris, France. [149]Pediatric endocrinology and diabetes, Hopital Necker Enfants Malades, APHP Centre, université de Paris, Paris, France. [150]Department of Medical Genetics, Haukeland University Hospital, Bergen, Norway. [151]Department of Medicine, University of Maryland School of Medicine, Baltimore, MD, USA. [152]Department of Epidemiology, Geisel School of Medicine at Dartmouth, Hanover, NH, USA. [153]Nephrology, Dialysis and Renal Transplant Unit, IRCCS—Azienda Ospedaliero-Universitaria di Bologna, Alma Mater Studiorum University of Bologna, Bologna, Italy. [154]Department of Medical Genetics, AP-HP Pitié-Salpêtrière Hospital, Sorbonne University, Paris, France. [155]Global Center for Asian Women's Health, Yong Loo Lin School of Medicine, National University of Singapore, Singapore, Singapore. [156]Department of Obstetrics and Gynecology, Yong Loo Lin School of Medicine, National University of Singapore, Singapore, Singapore. [157]Kaiser Permanente Northern California Division of Research, Oakland, CA, USA. [158]Department of Epidemiology and Biostatistics, University of California San Francisco, San Francisco, CA, USA. [159]National Institute of Diabetes and Digestive and Kidney Diseases, National Institutes of Health, Bethesda, MD, USA. [160]Department of Health Research Methods, Evidence, and Impact, Faculty of Health Sciences, McMaster University, Hamilton, ON, Canada. [161]Ann & Robert H. Lurie Children's Hospital of Chicago, Department of Pediatrics, Northwestern University Feinberg School of Medicine, Chicago, IL, USA. [162]Department of Clinical and Organizational Development, Chicago, IL, USA. [163]American Diabetes Association, Arlington, VA, USA. [164]College of Medicine and Health Sciences, University of Gondar, Gondar, Ethiopia. [165]Global Health Institute, Faculty of Medicine and Health Sciences, University of Antwerp, 2160 Antwerp, Belgium. [166]Department of Medicine and Kovler Diabetes Center, University of Chicago, Chicago, IL, USA. [167]School of Nursing, Faculty of Health Sciences, McMaster University, Hamilton, ON, Canada. [168]Division of Endocrinology, Metabolism, Diabetes, University of Colorado, Boulder, CO, USA. [169]Department of Clinical Medicine, School of Medicine, Trinity College Dublin, Dublin, Ireland. [170]Department of Endocrinology, Wexford General Hospital, Wexford, Ireland. [171]Division of Endocrinology, NorthShore University HealthSystem, Skokie, IL, USA. [172]Department of Medicine, Pritzker School of Medicine, University of Chicago, Chicago, IL, USA. [173]Department of Genetics, Stanford School of Medicine, Stanford University, Stanford, CA, USA. [174]Faculty of Health, Aarhus University, Aarhus, Denmark. [175]Departments of Pediatrics and Medicine and Kovler Diabetes Center, University of Chicago, Chicago, IL, USA. [176]University of Washington School of Medicine, Seattle, WA, USA. [177]Department of Population Medicine, Harvard Medical School, Harvard Pilgrim Health Care Institute, Boston, MA, USA. [178]Department of Medicine, Universite de Sherbrooke, Sherbrooke, QC, Canada. [179]Department of Internal Medicine, Seoul National University College of Medicine, Seoul National University Hospital, Seoul, Republic of Korea. [180]Joslin Diabetes Center, Harvard Medical School, Boston, MA, USA. [181]Charles Bronfman Institute for Personalized Medicine, Icahn School of Medicine at Mount Sinai, New York, NY, USA. [182]Broad Institute, Cambridge, MA, USA. [183]Division of Metabolism, Digestion and Reproduction, Imperial College London, London, UK. [184]Department of Diabetes & Endocrinology, Imperial College Healthcare NHS Trust, London, UK. [185]Department of Diabetology, Madras Diabetes Research Foundation & Dr. Mohan's Diabetes Specialities Centre, Chennai, India. [186]Department of Medicine, Faculty of Medicine and Health Sciences, University of Auckland, Auckland, New Zealand. [187]Auckland Diabetes Centre, Te Whatu Ora Health New Zealand, Auckland, New Zealand. [188]Medical Bariatric Service, Te Whatu Ora Counties, Health New Zealand, Auckland, New Zealand. [189]Oxford NIHR Biomedical Research Centre, University of Oxford, Oxford, UK. [190]University of Cambridge, Metabolic Research Laboratories and MRC Metabolic Diseases Unit, Wellcome-MRC Institute of

Metabolic Science, Cambridge, UK. [191]Department of Epidemiology & Public Health, University of Maryland School of Medicine, Baltimore, MD, USA. [192]Department of Internal Medicine, Division of Metabolism, Endocrinology and Diabetes, University of Michigan, Ann Arbor, MI, USA. [193]AdventHealth Translational Research Institute, Orlando, FL, USA. [194]Pennington Biomedical Research Center, Baton Rouge, LA, USA. [195]MRC Human Genetics Unit, Institute of Genetics and Cancer, University of Edinburgh, Edinburgh, UK. [196]Yale School of Medicine, New Haven, CT, USA. [197]Faculty of Medicine and Health, University of Sydney, Sydney, NSW, Australia. [198]Department of Endocrinology, Royal Prince Alfred Hospital, Sydney, NSW, Australia. [199]Kaiser Permanente Northwest, Kaiser Permanente Center for Health Research, Portland, OR, USA. [200]Clinial Research, Steno Diabetes Center Copenhagen, Herlev, Denmark. [201]Department of Clinical Medicine, Faculty of Health and Medical Sciences, University of Copenhagen, Copenhagen, Denmark. [202]Department of Endocrinology and Diabetology, University Hospital Düsseldorf, Heinrich Heine University Düsseldorf, Moorenstr. 5, 40225 Düsseldorf, Germany.

