## [Peer Review File · Communications Medicine]

Reviewers' comments:

Reviewer #1 (Remarks to the Author):

This is a systematic review of studies testing therapies modifying type 1 diabetes in relation to treatment response, aiming to assess current knowledge of precision approaches to T1D prevention in the research literature. The review describes the high quality of prevention and intervention studies but the low quality of precision analyses, which makes it difficult to draw meaningful conclusions, and calls for prespecified precision analyses to be included in the design of future studies.

The manuscript is well written and the topic is of particular interest to the reader. I have only two minor comments:

1. Table 1, row 2, FINNDIA study by Vaarala (2012), the intervention was performed using two whey-based formulations and only one of them (the FINDIA whey-based formulation essentially without bovine insulin) was positive, the HR in the table is not correct (not significant), please revise the table accordingly.
2. Supplementary Table 1, Age column - please add age range, not just child/adult/both categories.

Reviewer #2 (Remarks to the Author):

Review

Manuscript: Type 1 diabetes prevention: a systematic review (..) Felton et al.

This systematic review summarises the current evidence on prevention and intervention studies in type 1 diabetes in the context of precision medicine. It is written on behalf of ADA/EASD for a consensus report on precision diabetes medicine, the second edition. The review contains results of 75 articles, divided in studies on primary prevention (diabetes stage < 3), or treatments initiated after onset of diabetes (stage 3) aimed at prevention of ongoing betacell loss. The conclusion is that studies have high quality, in general, but lack in a major part robust and high quality precision analyses. This hampers application of the trial results for daily practice. Of course, the authors then recommend to incorporate prespecified precision analyses in future studies.

The subject is relevant, especially in T1D in which heterogeneity is increasingly recognized in all kinds of -omics and -types. It has originality as it is not a standard review on clinical or lab outcomes, but also reviews quality of papers in this specific context.

Methods:

the manuscript is based on a standard way of performing a systematic review according to standards. This is done appropriately. It is, however, not clear why (line 149) systematic reviews or meta-analyses of RCT's would add to the information needed. Given the context of precision analyses it is to be considered to only use the 'pure' results of studies and post-hoc analyses. A main concern is that the review covers 2 areas of pre- or early diabetes (namely stage < 3 and early stage 3) and thus combines different types of prevention (primary or secondary, and tertiary prevention after onset of disease). The authors do not explain this choice, and do not provide for the broader readership, the current view on the pathophysiology of T1DM (e.g. Insel 2015, Diab Care).

Throughout the article, it is difficult to separate what info is relevant for what kind of prevention. As an example the first lines of the conclusions (line 365-367) talk about ‘focused on disease modifying agents in T1D (..)’ while it also applies to the prevention in stages 1 or 2, which is generally not considered to be T1D already. I doubt about the need to include the studies on primary and secondary prevention. Probably the manuscript will gain expressiveness and ease of read when limiting to intervention studies. It will be helpful to use prevention for primary and secondary prevention, and intervention for any action after onset of disease.

Line 156: .. precision analyses... how is this defined, as it is a central issue in the manuscript

Line 160: ..addition key articles identified.. how?

Line 188: heterogeneity ..in the PRISMA checklist it is checked as NA: what does that mean? A short comment on the role of heterogeneity of T1DM pathophysiology (e.g. kids vs. adult, male vs. female) is essential as part of understanding the results of the review

Line 203: 1005 studies: in figure 1 (PRISMA flow diagram) it is 1006, please correct

Fig 2: provides data on primary and secondary prevention trials. See my comment above: do we need this?

Fig 3: 3a is not so informative. 3B: what is meant by ‘both’? Text in manuscript should be shortened when info can be found in figure. Again: no separation between prevention and intervention

Fig 4:

4C: primary trial papers: it provides 30 studies testes for age, but how many studies were included here? To better see the differences between all – precision – prim trial – f/u papers is can be considered to use percentages in this part of the figure, as well as in 4D

4D does not use the same parameters in each subfigure, so difficult to read/ interpret. C peptide is not further specified on random sample, or MMTT AUC... please explain that in legenda.

Conclusions

This section is very long, and more a discussion than conclusion part. Please condense and focus on explaining and interpreting the review findings (‘what did we learn..’) instead of going into several studies. Line 491-493 is not relevant, because logical. I would appreciate the authors to provide on a list of improvements for future studies, based on this review, as support for the clinicians developing new studies.

Abstract

Line 49 .. prevention... but also intervention

Title

Should be changed depending on what will be provided in the article, using prevention only does not cover the content now

Reviewer #3 (Remarks to the Author):

Reviewer’s opinion

Thanks for the efforts of Jamie L. Felton et al. This study entitled “Type 1 Diabetes Prevention: a systematic review of studies testing disease-modifying therapies and features linked to treatment

response” presented a comprehensive systematic review to address the un-elucidated question that whether disease modifying therapies (including immunomodulatory therapies, and pathology alleviation therapies such as beta cell health preserving agents) are effective in type 1 diabetes (T1D) prevention. The authors retrieved available previous randomized controlled trials, reviewed the effects of different types of therapy in primary or secondary T1D prevention, and conducted subgroup analyses investigating the impact of potential influencing features (as age, sex and so on) on therapy response. The quality of included studies was high, and the authors provided several clinical insights in this review. While the authors also mentioned that the low quality of precision analyses made the conclusion difficult to instruct clinical practice, and further prespecified precision analyses should be incorporated into the design of future studies to settle this problem.

Overall, the research topic is novel and of clinical significance, with plain and concise languages. There are no missing items and processes in study retrieval, inclusion and exclusion standards, data synthesis & analyses, and literature quality assessments. The inclusion of relevant studies is also up to date. It’s highly anticipated that this work would complement the resources of evidence-based medicine in T1D prevention, and provide inspirations and references for future researches.

However, there are still some questions and advice for this article to note here:

1. The main question confused me is that what is the clinical significance of your study. What does your study add to the current literature?
2. In introduction part, it is recommended to add more elucidations for what exactly the disease modifying therapies for T1D are, and what specific types of therapy they mainly include.
3. In conclusion part, there are actually more limitations need to be addressed, such as the heterogeneity of included studies, and the low quality of precision analyses with absent prespecified precision analyses. Please review the limitations section and detail each.
4. In conclusion part, please amplify the clinical significance of your study in the summary section. The texts now are more concentrated on future directions and the drawbacks of your study, while the illustrations for what the clinical significance is, and what does your study add to the current literature are still in lack.

Point by point response to reviewers

Reviewer #1 (Remarks to the Author):

This is a systematic review of studies testing therapies modifying type 1 diabetes in relation to treatment response, aiming to assess current knowledge of precision approaches to T1D prevention in the research literature. The review describes the high quality of prevention and intervention studies but the low quality of precision analyses, which makes it difficult to draw meaningful conclusions, and calls for prespecified precision analyses to be included in the design of future studies. The manuscript is well written and the topic is of particular interest to the reader. I have only two minor comments:

We thank Reviewer for providing this critique, as well as acknowledgment of the timeliness of this review topic.

Table 1, row 2, FINNDIA study by Vaarala (2012), the intervention was performed using two whey-based formulations and only one of them (the FINDIA whey-based formulation essentially without bovine insulin) was positive, the HR in the table is not correct (not significant), please revise the table accordingly.

We thank the reviewer for the close reading of this manuscript and have revised and updated Table 1 to now include both interventions with both hazard ratios, consistent with Figure 1.

Supplementary Table 1, Age column - please add age range, not just child/adult/both categories.

We agree that this addition enriches the information presented and in response to this suggestion have replaced the child/adult category column with age ranges in Supplementary Table 1.

Reviewer #2 (Remarks to the Author):

This systematic review summarises the current evidence on prevention and intervention studies in type 1 diabetes in the context of precision medicine. It is written on behalf of ADA/EASD for a consensus report on precision diabetes medicine, the second edition. The review contains results of 75 articles, divided in studies on primary prevention (diabetes stage < 3), or treatments initiated after onset of diabetes (stage 3) aimed at prevention of ongoing beta cell loss. The conclusion is that studies have high quality, in general, but lack in a major part robust and high quality precision analyses. This hampers application of the trial results for daily practice. Of course, the authors then recommend to incorporate prespecified precision analyses in future studies. The subject is relevant, especially in T1D in which heterogeneity is increasingly recognized in all kinds of -omics and -types. It has originality as it is not a standard review on clinical or lab outcomes, but also reviews quality of papers in this specific context.

We thank Reviewer 2 for the thoughtful comments provided and recognition of the relevance of the current review in the context of the field.

Methods:

the manuscript is based on a standard way of performing a systematic review according to standards. This is done appropriately. It is, however, not clear why (line 149) systematic reviews or meta-analyses of RCT's would add to the information needed. Given the context of precision analyses it is to be considered to only use the 'pure' results of studies and post-hoc analyses.

Meta-analyses and systematic reviews were initially included in the search strategy to ensure that papers combining pre-existing studies to provide better powered precision analyses were not missed. However, in reality, no meta-analyses or systematic reviews met full inclusion criteria, and so none are included in our systematic review. We agree with the reviewer that this is an important point to highlight, and this is now noted in the manuscript (lines 167-170).

A main concern is that the review covers 2 areas of pre- or early diabetes (namely stage < 3 and early stage 3) and thus combines different types of prevention (primary or secondary, and tertiary prevention after onset of disease). The authors do not explain this choice, and do not provide for the broader readership, the current view on the pathophysiology of T1DM (e.g. Insel 2015, Diab Care). Throughout the article, it is difficult to separate what info is relevant for what kind of prevention. As an example the first lines of the conclusions (line 365-367) talk about 'focused on disease modifying agents in T1D (..)' while it also applies to the prevention in stages 1 or 2, which is generally not considered to be T1D already. I doubt about the need to include the studies on primary and secondary prevention. Probably the manuscript will gain expressiveness and ease of read when limiting to intervention studies. It will be helpful to use prevention for primary and secondary prevention, and intervention for any action after onset of disease.

As part of the larger Precision Medicine in Diabetes Initiative, we were tasked to systematically review literature focused on T1D prevention, which was defined in the first ADA/EASD Precision Medicine in Diabetes Consensus Report as "using information about a person's unique biology, environment, and/or context to determine their likely responses to health interventions" and states that "precision prevention should optimize the prescription of health-enhancing interventions" (lines 108-112). As noted by the reviewer, T1D development occurs along a spectrum of progressive beta cell loss. However, this spectrum does not begin or end with a clinical type 1 diabetes diagnosis. Instead, beta cell damage and loss predate clinical diagnosis and continues to progress after diagnosis, with most individuals continuing to secrete insulin during the period immediately following diagnosis. Because preservation of endogenous insulin production at any point along this spectrum, including immediately following a clinical type 1 diabetes diagnosis, has been shown to be beneficial for long-term management of T1D, we chose to include studies focusing on disease-modifying therapies both before and around the time of a clinical type 1 diabetes diagnosis. Furthermore, because most agents that would ideally or will ultimately target T1D prevention are first tested in the recent-onset period, we felt that a "prevention" review that did not include "intervention" studies would lack critical data needed for comprehensive review and conclusions on the state of the data surrounding this topic.

We agree that the manuscript will benefit from clarification of these concepts and thank the reviewer for the insight into ways to improve expressiveness and readability for a broader readership. Description of T1D pathophysiology and justification for inclusion of studies before and after diagnosis have been added to the abstract (lines 42-43) and introduction (lines 126-141). Clarification has been added to specify that "disease modifying agents" refer to those that delay/prevent clinical diagnosis or slow decline in endogenous beta cell function after diagnosis (lines 91-94, 128-130). We also clarified definitions of "prevention" and "new-onset" studies (lines 133-137).

Because the spectrum of beta cell loss is so broad and can span decades in some individuals, we felt that it is important to distinguish between studies in populations without established autoimmunity (primary prevention) from those with established autoimmunity and often metabolic derangements (secondary prevention). We have clarified the definitions of primary and secondary prevention trials (lines 217-222).

Because many of the trials included were conducted prior to publication of the staging system, we were unable to utilize the Insel 2015 staging system terminology throughout the paper. However, we introduced this concept in the new intro text describing the spectrum of disease (lines 121-130) and have included limited discussion of stages per the Insel 2015 paper to the conclusions (lines 486-503).

Line 156: .. precision analyses... how is this defined, as it is a central issue in the manuscript

Precision analysis refers to the analysis of specific individual measures or features of response to disease modifying agents. This has been clarified in the text (lines 161-162).

Line 160: ..addition key articles identified.. how?

Given that our group of authors included established experts in the field, additional articles previously known to the authors as contributing to the field but not identified as part of the original search strategy were also included if they otherwise met the prespecified article inclusion criteria. This has been clarified in the text (lines 164-167).

Line 188: heterogeneity ..in the PRISMA checklist it is checked as NA: what does that mean? A short comment on the role of heterogeneity of T1DM pathophysiology (e.g. kids vs. adult, male vs. female) is essential as part of understanding the results of the review

Line 13e in the PRISMA checklist is marked n/a because combined subgroup analyses and/or meta-analyses were not possible as part of this review due to major differences in treatments applied, participant populations studied, and reporting of outcome measures for trials. We agree with the reviewer that clinical heterogeneity is a key challenge to highlight in the T1D prevention space. We have highlighted this, and the implications for trial design in the discussion (lines 385-395).

Line 203: 1005 studies: in figure 1 (PRISMA flow diagram) it is 1006, please correct

We thank the reviewer for their attention to this detail. This has been corrected.

Fig 2: provides data on primary and secondary prevention trials. See my comment above: do we need this?

As mentioned above, after careful consideration we have respectfully decided not to exclude primary and secondary prevention trials, but have instead have added definitions and clarifications throughout the manuscript to improve readability (please see response above).

Fig 3: 3a is not so informative. 3B: what is meant by "both"? Text in manuscript should be shortened when info can be found in figure. Again: no separation between prevention and intervention

Based on the reviewer's feedback on Figure 3a we have removed this figure from the manuscript. In Figure 3B, "both" describes papers that included both pre-specified and post-hoc analyses. This has now been clarified in the figure legend.

Fig 4:

4C: primary trial papers: it provides 30 studies testes for age, but how many studies were included here? To better see the differences between all – precision – prim trial – f/u papers is can be considered to use percentages in this part of the figure, as well as in 4D

We thank the reviewer for this opportunity for clarification. This figure is inclusive of all papers that included precision analyses. We have now included sample sizes in each panel of this figure for clarity. We considered using percentages for these images, but because many papers studied included more than one feature or outcome, total percentages added up to >100 and we thought this would be confusing for readers. Instead, we have chosen to utilize absolute numbers for each feature studied, with clarified total numbers for each paper type.

4D does not use the same parameters in each subfigure, so difficult to read/ interpret. C peptide is not further specified on random sample, or MMTT AUC... please explain that in legend.

To allow for larger figure text, we previously did not include parameters for which zero papers in the category being displayed tested the feature or used the endpoint. However, in response to this suggestion, we have now included all features or outcomes analyzed within any of the paper categories in each panel. The C-peptide measure category was inclusive of any measure of beta cell function, including mixed meal area under the curve, stimulated C-peptide values, fasting C-peptide values, etc. However, specific metabolic endpoints utilized for each study are listed in more detail in Supplemental Table 2.

Conclusions

This section is very long, and more a discussion than conclusion part. Please condense and focus on explaining and interpreting the review findings ('what did we learn..') instead of going into several studies. Line 491-493 is not relevant, because logical. I would appreciate the authors to provide on a list of improvements for future studies, based on this review, as support for the clinicians developing new studies.

Thank you for this feedback. We considered this feedback and comments from Reviewer 3 and reworked our conclusions based on these comments. We felt it was important to highlight key studies that may not have met our inclusion criteria but could theoretically contribute to the literature on T1D disease-modification, as well as to highlight some of the biological findings emanating from our review, but also key takeaways as suggested by both reviewers. In response to this comment and to address the comments from Reviewer 3, the conclusion section has been cut down, and revised to highlight the main take-home messages that were drawn from this review.

Abstract

Line 49 .. prevention... but also intervention

We have modified this line to read disease-modification and elaborated (as noted in responses above) in the text regarding disease modification along the spectrum of beta cell destruction.

Title

Should be changed depending on what will be provided in the article, using prevention only does not cover the content now

We agree with the reviewer's point that using prevention only in the title requires clarification in the text, based on the studies included. As part of the larger Precision Medicine in Diabetes initiative, our charge is to serve as the "prevention" group; thus, we are unable to change the title to read prevention and intervention. However, to address this point we have defined and justified why we included studies aimed at disease modification both before and immediately following clinical T1D diagnosis. We have also included "disease-modifying therapies" in the title to be inclusive of studies after disease has been established.

Reviewer #3 (Remarks to the Author):

Thanks for the efforts of Jamie L. Felton et al. This study entitled "Type 1 Diabetes Prevention: a systematic review of studies testing disease-modifying therapies and features linked to treatment response" presented a comprehensive systematic review to address the un-elucidated question that whether disease modifying therapies (including immunomodulatory therapies, and pathology alleviation therapies such as beta cell health preserving agents) are effective in type 1 diabetes (T1D) prevention. The authors retrieved available previous randomized controlled trials, reviewed the effects of different types of therapy in primary or secondary T1D prevention, and conducted subgroup analyses investigating the impact of potential influencing features (as age, sex and so on) on therapy response. The quality of included studies was high, and the authors provided several clinical insights in this review. While the authors also mentioned that the low quality of precision analyses made the conclusion difficult to instruct clinical practice, and further prespecified precision analyses should be incorporated into the design of future studies to settle this problem.

Overall, the research topic is novel and of clinical significance, with plain and concise languages. There are no missing items and processes in study retrieval, inclusion and exclusion standards, data synthesis & analyses, and literature quality assessments. The inclusion of relevant studies is also up to date. It's highly anticipated that this work would complement the resources of evidence-based medicine in T1D prevention, and provide inspirations and references for future researches.

We thank Reviewer 3 for the constructive review, as well as the recognition of the relevance and novelty of our work. We also thank the reviewer for application of his/her expertise in systematic reviews and acknowledgement of the quality of our methodology in the current review.

However, there are still some questions and advice for this article to note here:

1. The main question confused me is that what is the clinical significance of your study. What does your study add to the current literature?

We appreciate the opportunity to clarify and strengthen the overall impact of this manuscript. We have now revised the discussion section to emphasize our key take home points: studies are high quality but precision analyses are not high enough quality to support the existence of phenotypes linked to

treatment response. We go on to highlight key recurring points that should be addressed to move forward precision medicine in T1D prevention: 1) Standardization of approaches to outcomes for precision analyses; 2) Consideration of a recurring role for age and measures of beta cell function; 3) Reproducible biomarkers linked to underlying disease pathology; 4) A need for pre-specified, appropriately powered precision analyses; and 5) consideration of impacts of the T1D staging system. The Conclusions section has been significantly revised to reflect and highlight these points.

2. In introduction part, it is recommended to add more elucidations for what exactly the disease modifying therapies for T1D are, and what specific types of therapy they mainly include.

We agree with the reviewer's point. In response to this comment, the introduction has been revised to include a more thorough discussion of the spectrum of disease progression and the goals of disease-modifying therapies throughout this progression (lines 91-99).

3. In conclusion part, there are actually more limitations need to be addressed, such as the heterogeneity of included studies, and the low quality of precision analyses with absent prespecified precision analyses. Please review the limitations section and detail each.

We agree that these limitations need to be addressed. Because several of the limitations described above are also central to the key take-home points (heterogeneity of studies and low quality precision analyses), clarifications and details of these limitations have been added within those sections of the conclusion. However, in response to this comment, we added heterogeneity of papers studies (limiting ability for meta-analysis) and the lack of diversity in populations studied to the limitations section (lines 506-517).

4. In conclusion part, please amplify the clinical significance of your study in the summary section. The texts now are more concentrated on future directions and the drawbacks of your study, while the illustrations for what the clinical significance is, and what does your study add to the current literature are still in lack.

This is an excellent point and in response, the conclusion section of the manuscript has been significantly revised. We start with highlights for clinical significance, followed by a focus on the main obstacles that need to be addressed in order to translate precision medicine into clinical T1D prevention forward.

REVIEWERS' COMMENTS:

Reviewer #1 (Remarks to the Author):

I do not have any additional comments.

Reviewer #3 (Remarks to the Author):

Thanks for the authors' careful and significant response. The completeness and readability of the revised manuscript are greatly improved since the detailed demonstrations of several previously missed points have been adequately replenished.

With the extensions of introduction for disease modifying therapies in type 1 diabetes, the elicitation of the following study aims and scientific questions became more rational. The objectivity and strictness of this work have also been enhanced through fully listing the limitations and expounding ways adopted to control or elude them. Furthermore, by adding more explanations for the clinical significance of this research, and rearranging the narrative sequences, this research exhibited a comprehensive story about the raising questions, conducting researches, obtaining results and presenting clinical significance & future directions.

Overall, the revised manuscript is well-done and adequately answered my preceding questions.